# Real word evidence on rituximab utilization: Combining administrative and hospital-pharmacy data

Giuseppe Roberto[1☯], Andrea Spini[2☯], Claudia Bartolini[1], Valentino Moscatelli[3], Alessandro Barchielli[4], Davide Paoletti[5], Silvano Giorgi[5], Alberto Fabbri[6], Monica Bocchia[6], Sandra Donnini[3], Rosa Gini[1], Marina Ziche[2]*

**1** Agenzia Regionale di Sanità della Toscana, Firenze, Italy, **2** Department of Medical Science, Surgery and Neuroscience, University of Siena, and Azienda Ospedaliera Universitaria Senese, Siena, Italy, **3** Department of Life sciences, University of Siena, Siena, Italy, **4** Istituto per lo Studio la Prevenzione e la Rete Oncologica, Firenze, Italy, **5** Farmacia Oncologica, Azienda Ospedaliera Universitaria Senese, Siena, Italy, **6** Unit of Hematology, Azienda Ospedaliera Universitaria Senese and University of Siena, Siena, Italy

☯ These authors contributed equally to this work.
* marina.ziche@unsi.it

**Data Availability Statement:** All relevant data are within the manuscript and its Supporting Information files.

## Abstract

### Purpose

To describe patterns of utilization, survival and infectious events in patients treated with rituximab at the University Hospital of Siena (UHS) to explore the feasibility of combining routinely collected administrative and hospital-pharmacy data for examining the real-world use of intravenous antineoplastic drugs.

### Methods

A retrospective, longitudinal cohort study was conducted using data from the Hospital Pharmacy of Siena (HPS) and the Regional Administrative Database of Tuscany (RAD). Patients aged ≥18 years with ≥1 rituximab administration recorded between January 2012 and June 2016 were identified in the HPS database. Anonymized patient-level data were linked to RAD. Rituximab utilization during the first year of treatment was described using HPS. Hospital diagnoses of adverse infectious events that occurred during the first year of follow-up and four-year survival were observed using RAD.

### Results

A total of 311 new users of rituximab were identified: 264 patients received rituximab for non-Hodgkin's lymphoma (NHL) and 47 were treated for chronic lymphocytic leukemia (CLL). Among new users with one complete year of follow-up (n = 203) over 95% received rituximab as the first-line treatment, and approximately 70% of them received 5–8 doses. No patient in the CLL group received >8 administrations. Four-year survival was approximately 70% in both CLL and NHL patients. Sepsis was the most frequent infectious event observed (5.1%).

**Funding:** The funders had no role in study design, data collection and analysis, decision to publish, or preparation of the manuscript.

**Competing interests:** The authors have declared that no competing interests exist.

## Conclusion

HPS and RAD provided complementary information on rituximab utilization, demonstrating their potential for future pharmacoepidemiological studies on antineoplastic medications administered in the Italian hospital setting. Overall, this general description of the real-world utilization of rituximab in patients treated for NHL and CLL at UHS was in line with treatment guidelines and current knowledge on the rituximab safety profile.

## Introduction

Rituximab is a monoclonal antibody first approved by the European Medicines Agency in 1998. It is currently licensed for the treatment of the adult with non-Hodgkin's lymphoma (NHL), chronic lymphocytic leukemia (CLL) rheumatoid arthritis, granulomatosis with poly-angiitis and microscopic polyangiitis.[1–5] The therapeutic efficacy of rituximab is based on the B-cell depletion mediated by the selective binding of CD20 receptors expressed on the surface of these cells.[1] By causing the depletion of B lymphocytes, rituximab interferes with humoral immunity, and the risk of infections represent one of the major safety concerns associated with its use. In particular, treatment with rituximab has been associated with episodes of bacteremia, sepsis, and other opportunistic infections, including reactivation of latent viral infections. [1–2,6]

As for the hematological malignancies, with the exception of early stage follicular lymphoma, rituximab is considered the first-line treatment for patients with NHL and CLL.[1,7] In newly treated patients rituximab is administered in combination with chemotherapy. Monotherapy with rituximab is indicated as maintenance therapy for patients who responded to induction therapy. Monotherapy is also indicated in patients with stage III-IV follicular lymphoma who are chemo-resistant or those in their second or subsequent relapse after chemotherapy. In particular, the administration of 375 mg/m2 is recommended, for both induction and maintenance therapy, for the treatment of NHL, while the total number of administrations may range from 4 to 12 depending on the specific subtype of NHL to be treated (i.e. follicular lymphoma or diffuse large B cell). On the other hand, the recommended dose of rituximab for treating CLL is 375 mg/m2 for the first administration followed by 500 mg/m2 for subsequent cycles, for a total of 6 cycles.[1]

To date, real-world evidence on rituximab utilization in the large adult population affected by onco-hematological malignancies is still poor. [8] In Italy, electronic administrative databases, which are the most widely used data source for studying drug utilization in large populations [9,10] cannot accurately track the use of drugs in the hospital setting. However, the missing information about the drug use, such as the indication of use, treatment line, and drug administration during inpatient care, can be found in other existing intra-hospital databases like the hospital pharmacy management system database.

The aim of this study was to explore the feasibility of linking records from the electronic database of the Hospital Pharmacy of Siena (HPS) and the Regional Administrative Database of Tuscany (RAD) to generate real-world evidence on intravenous antineoplastic drug use in Italy. With this purpose, we described the pattern of drug use, survival and hospital admissions for adverse infectious events, in patients treated with rituximab for hematologic malignancies at the University Hospital of Siena (UHS).

## Methods

### Data sources

RAD collects information on healthcare services dispensed to Tuscan inhabitants and reimbursed by the National Healthcare Service. [9–11] RAD includes different registries that can be linked through a pseudo-anonymized regional person identifier code (ID). For this study we used RAD data from the following three sources: i) the population registry, which records demographic information, including vital status (if the patients were alive or dead) of all inhabitants entitled to public healthcare assistance; ii) the outpatient's drug dispensing registry, which include information on reimbursed drugs for outpatient use (i.e. drug name, formulation, date of dispensing and the Anatomical Therapeutic Chemical (ATC) Classification System code), and iii) the hospital discharge records, which included inpatient diagnoses and procedures associated to hospitalization events.

HPS database records information concerning intravenous drugs prepared by the hospital pharmacy and administered to both inpatients and outpatients treated at the UHS. The database contains demographic data of treated patients as well as information on drug administration, including the indication of use, date of administration, treatment line and dose.

### Dataset creation

The competent regional authority (ESTAR—Ente di Supporto Tecnico-Amministrativo Regionale) attributed a pseudo-anonymized ID to all patients identified in HPS as having ≥1 record of rituximab treatment recorded between January 1, 2009, and June 30, 2016, in the oncology or hematology unit of the AOUS (Azienda Ospedaliera Universitaria Senese). The anonymized data from HPS were then linked to RAD using the regional ID.

### Population selection, study design, and statistical analysis

We described utilization of rituximab for patients who received at least 1 dose of the drug between Jan 1, 2009, and June 30, 2016, as 1st line treatment for CLL and NHL. The first rituximab administration recorded in HPS between January 1, 2012, and June 30, 2016, was the *cohort entry date*. Patients aged ≤18 at cohort entry were excluded as well as those for which the linkage to RAD was not feasible. Furthermore, patients with ≤1 year of the *look-back period* (i.e. the observation time that runs backward with respect to cohort entry [12]) in RAD were excluded. Patients of Tuscany admitted to hospitals other than UHS were also excluded. Finally, we excluded from the analysis all the patients with indication other than CLL and NHL

*New users* of rituximab (i.e. patients who received rituximab administration for the first time [13]) were identified and defined as patients with no records of rituximab administration or dispensing neither in HPS nor in RAD before the *index date*.

Patients were classified as NHL or CLL patients on the basis of the indication of use recorded in the hospital pharmacy database in association with the first rituximab administration (i.e. at cohort entry). The one-year mortality rate was calculated by indication of rituximab use, as recorded in HPS, and the survival curves were plotted applying the Kaplan-Meier method to all the available follow-up time data in RAD (30 June 2016).

To describe the utilization pattern of rituximab during the first year of treatment, new users with 1 complete year of follow-up in RAD were selected. Using HPS data, age, sex, treatment line, number of rituximab administrations and mean dose received during one year were described.

Adverse infectious events occurring during the first year of treatment were also described considering all new users, regardless of minimum follow-up duration. Adverse infectious

events were described in terms of type and frequency using diagnoses recorded in hospital discharge records available from RAD. The list of infectious events proposed by Kavicic et al, [14] with corresponding ICD9CM codes, was applied. The distribution of the time-to-onset of the observed infectious events from the first rituximab administration was also described.

Since NHL and CLL are different diseases and guidelines governing rituximab use change between these pathologies, all the analyses described above were executed and reported according to the indication of rituximab use recorded in HPS, i.e. NHL and CLL respectively. The study was approved by the Ethical Committees section "Area Vasta Sud-Est" in July 2016 (code identifier: ARSAOUS2016).

## Results

A total of 425 users of rituximab were identified in HPS. For approximately 90% of these patients (N = 392), data recorded in HPS could be linked to those in RAD. (Fig 1)

A total of 311 new users were identified, corresponding to 264 patients with NHL and 47 with CLL.

The one-year mortality rate was 7.8 per 100 person-months for CLL and 6.9 per 100 person-months for NHL. The 4-years overall survival (OS) rates for both CLL and NHL patients

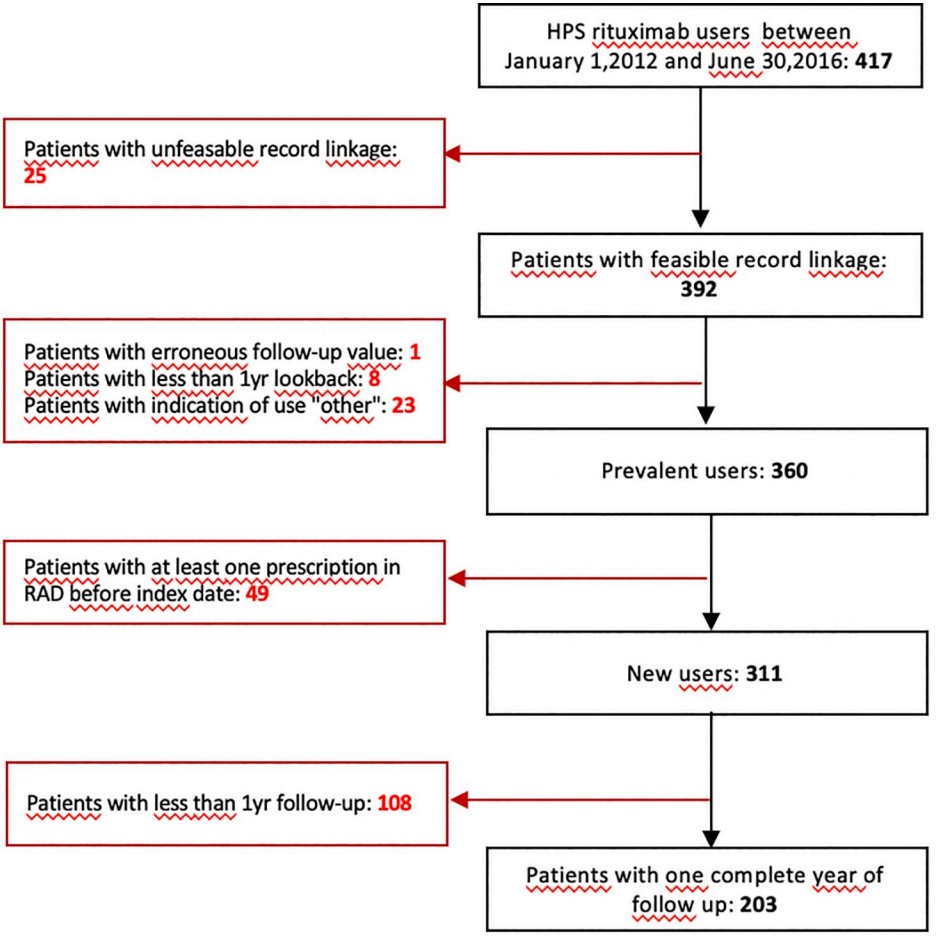

**Fig 1. Study flow chart.** The picture represents the number of rituximab users identified in HPS as prevalent and new users. All exclusion motivation were reported.

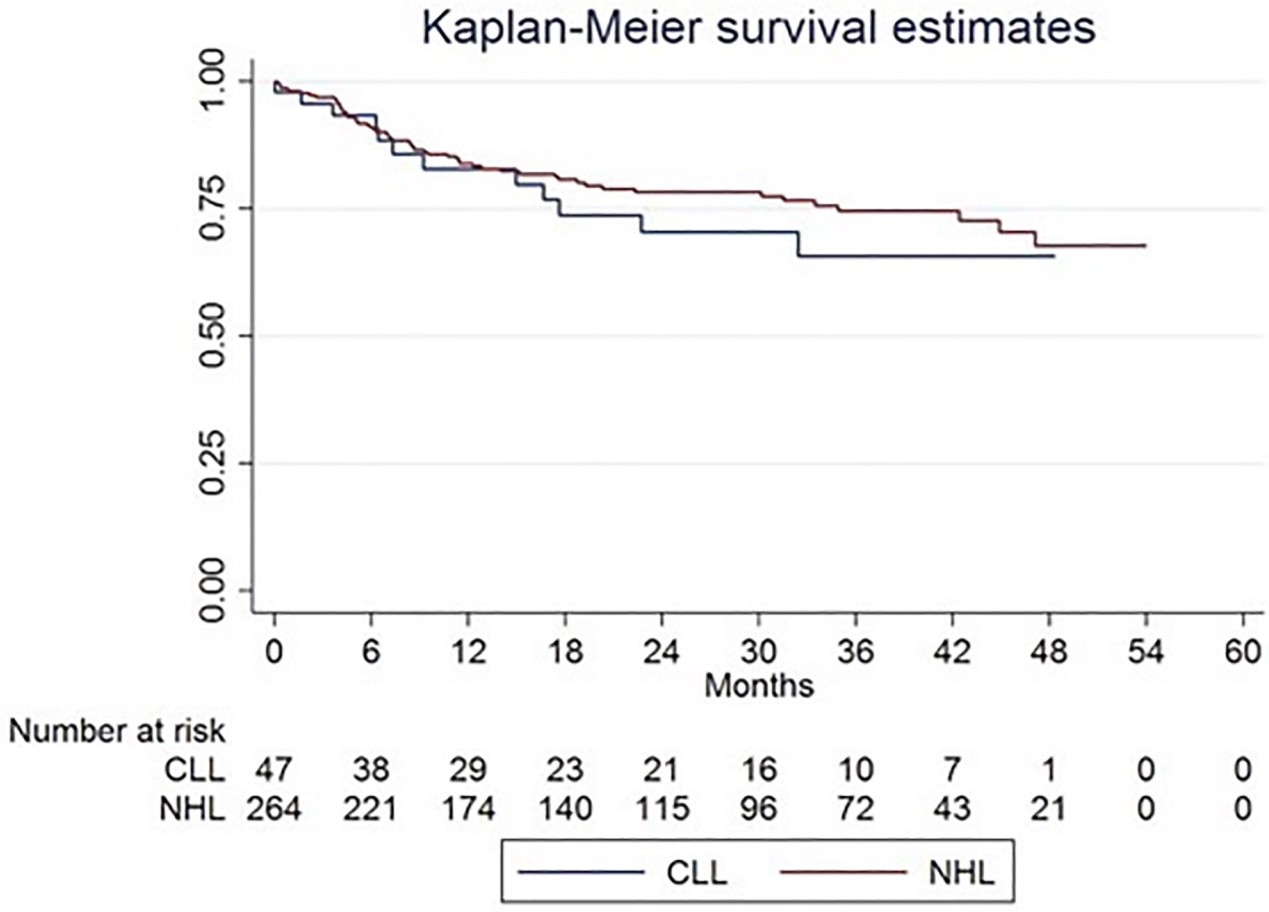

**Fig 2. Kaplan Meier overall survival analysis for CLL and NHL incident patients.** The picture represents the survival of incident patients with CLL (blue line) and NHL (red line). NHL: non-Hodgkin's lymphoma; CLL: chronic lymphocytic leukemia.

were approximately 70%. There was no significant difference between survival curves plotted for patients with NHL and CLL, respectively (log-rank test: p = 0.41). (Fig 2)

On a total of 311 new users of rituximab, 34.7% had < 1 year of follow-up (Table 1). In particular, during the first year of treatment 14.8% of them died, 0.6% emigrated from Tuscany,

**Table 1. Patient new users enrolled at the AOUS for treatment with rituximab.**

|  | Indications | | |
|---|---|---|---|
|  | **NHL** | **CLL** | **Total** |
| Overall new users (total) | 264 | 47 | 311 |
| New users with <1 year of follow-up, n (%) | 90 (34.1) | 18 (38.3) | 108 (34.7) |
| Death | 39 (14.7) | 7 (14.9) | 46 (14.8) |
| Emigration | 2 (0.8) | - | 2 (0.6) |
| Admission to a different hospital | 2 (0.8) | 1 (2.1) | 3 (0.9) |
| End of the study period | 47 (17.8) | 10 (21.3) | 57 (18.3) |
| New users with one year of follow up, n(%) | 174 (70.4) | 29 (61.7) | 203 (65.3) |

NHL: Non-Hodgkin Lymphoma; CLL: Chronic Lymphocytic Leukemia

**Table 2. Rituximab new users characteristics with at least one year of follow-up.**

| | Indications | | |
| --- | --- | --- | --- |
| | NHL (n = 174) | CLL (n = 29) | Total (n = 203) |
| Men/women ratio | 1.1 | 2.6 | 1.2 |
| Mean age | 65.2 | 69.9 | 63.9 |
| Women | 67.1 | 66. | 64.2 |
| Men | 63.4 | 71.2 | 63.7 |
| Age bands, n (%) | | | |
| 18–44 | 12 (7.5) | - | 12 (5.9) |
| 45–64 | 57 (32.7) | 9 (31) | 66 (32.5) |
| 65+ | 105 (60.4) | 20 (68.9) | 125 (61.6) |
| N. administrations per patient | | | |
| Mean (range) | 6.5 (1–11) | 5.7 (1–8) | 6.4 (1–11) |
| Distribution of patients per number of administrations, n (%) | | | |
| 1–2 | 9 (5.2) | 1 (3.5) | 10 (4.9) |
| 3–4 | 27 (15.5) | 4 (13.8) | 31 (15.3) |
| 5–6 | 42 (24.1) | 19 (65.5) | 61 (30.0) |
| 7–8 | 79 (45.4) | 5 (17.3) | 84 (41.4) |
| 8+ | 17 (9.7) | - | 17 (8.4) |
| Mean dose, mg | 659.2 | 788.2 | 675.7 |
| 1st dose | 648.9 | 675.8 | 652.7 |
| Doses after the 1st | 661.1 | 811.8 | 679.9 |
| Treatment line | | | |
| 1a | 168 (96.5) | 26 (89.6) | 194 (95.6) |
| 2a | 6 (3.5) | 3 (10.3) | 9 (4.4) |

NHL: Non-Hodgkin Lymphoma; CLL: Chronic Lymphocytic Leukemia

0.9% was admitted to a different hospital and 18.3% reached the end of the study period before the completion of one year of observation.

Among 203 new users with 1 complete year of follow-up, 174 received rituximab for the treatment of NHL and 29 for CLL. The ratio between men and women was 1.1 for NHL and 2.6 for CLL, and new users with ≥65 years old were 60.4% and 68.9% respectively (Table 2).

Overall, 95.6% of patients received rituximab as first-line treatment. As for the number of administrations received during the first year of treatment, 45.4% of patients with NHL received from 7 to 8 and 9.7% received > 8 doses. The majority of patients with CLL (65.5%) received from 5 to 6 doses, while only 5 patients (17.3%) received between 7 and 8 doses. No patients with CLL received > 8 doses. The widest difference between the mean dose at the first and the subsequent administrations was observed in users with CLL (1st dose 675.8 mg vs subsequent doses: 811.8 mg) (Table 2).

Among 311 new users, 23 (7%) had at least one hospital diagnosis of infectious disease recorded during the first year of treatment with rituximab. The most frequently diagnosed event was "sepsis" (n = 16) (Table 3). Most of the diagnoses considered for this analysis involved admissions to the UHS (n = 21/23). Notably, two patients had a record of viral hepatitis infection though they were not listed as the primary cause of hospital admission (S1 Table).

Finally, we described the time-to-onset of adverse infections events (S2 Table). The average time between the first rituximab administration and the first infectious events was lower for adenovirus infectious (62.2 days) compared to sepsis (131.7 days). We observed only one case of bacteremia and one of tuberculosis: the average time-to-onset was 9 and 203 days

Table 3. New users with at least one hospital diagnosis of adverse infectious events.

| | | Indication of use | | |
|---|---|---|---|---|
| | ICD9CM codes | NHL (n = 264) | CLL (n = 47) | Total (n = 311) |
| Patients with ≥1 adverse infectious event (from first rituximab administration up to 1 year) | | 20 (7.6) | 3 (6.4) | 23 (7.4) |
| Women (%) | | 5 (1.9) | 0 | 5 (1.6) |
| Admitted to University Hospital of Siena (%) | | 19 (7.2) | 2 (4.2) | 21 (6.7) |
| **Bacteremia** | 790.7 | **1 (0.4)** | **0** | **1 (0.3)** |
| **Sepsis** | | **15 (5.7)** | **1 (2.1)** | **16 (5.1)** |
| Staphylococcus aureus septicemia | 038.11 | 1 (0.4) | 0 | 1 (0.3) |
| Other staphylococcal septicemia | 038.19 | 2 (0.8) | 1 (2.1) | 3 (1) |
| Septicemia due to other gram-negative organisms | 038.4* | 3 (1.1) | 0 | 3 (1) |
| Unspecified septicemia | 038.9 | 3 (1.1) | 0 | 3 (1) |
| Systemic inflammatory response syndrome due to infectious process without organ dysfunction | 995.91 | 4 (1.5) | 0 | 4 (1.3) |
| Septic shock | 785.52 | 2 (0.8) | 0 | 2 (0.6) |
| **Adenovirus infection** | | **4 (1.5)** | **1 (2.1)** | **5 (1.6)** |
| Cytomegalovirus | 785 | 1 (0.4) | 0 | 1 (0.3) |
| Herpes simplex virus | 054.XX* | 0 | 1 (2.1) | 1 (0.3) |
| Herpes zoster (Human Herpesvirus 3) | 053.XX* | 1 (0.5) | 0 | 1 (0.3) |
| Hepatitis B virus | 070.3* | 2 (1.1) | 0 | 2 (0.6) |
| **Pulmonary tuberculosis (Mycobacterium tuberculosis)** | **1173** | **0** | **1 (2.1)** | **1 (0.3)** |

*Infections events, recorded during one year following the first administration of rituximab, were retrieved from diagnosis codes recorded in both primary or secondary position of the discharge record. Only the first documented infectious events per patients was described. NHL: Non-Hodgkin Lymphoma; CLL: Chronic Lymphocytic Leukemia

respectively (see S3 Table with the time-to-onset analysis restricted to the primary cause of hospital admission only).

## Discussion

To the best of our knowledge, this is the first study providing real-world evidence on rituximab utilization in the Italian clinical practice by combining information from routinely collected administrative data and a hospital pharmacy database. The analysis of the two data sources provided a general description of the utilization pattern, survival and adverse infectious events in patients treated with rituximab for NHL and CLL at the UHS. Overall, findings from this study were in line with current treatment recommendations and available knowledge on the rituximab safety profile. [15,16]

As expected from clinical guidelines, [17] almost all patients in the study cohort were recorded into the HPS database as receiving rituximab as first-line treatment for NHL and CLL. This was confirmed by the overlap of the observed distribution by sex and age of new users at the time of the first administration with the respective hematological pathologies for which the drug was used. [18,19]

The observed number of administrations during the first year treatment with rituximab, as well as the average administered dose, appeared to reflects the recommended posology for the treatment of NHL and CLL, as reported in the rituximab Summary of Product Characteristics (SmPC). [1] In particular, the majority of patients with CLL received up to 6 administrations during the first year of treatment. Unexpectedly, one out of 6 patients with CLL received between 7 and 8 doses. Although these findings might be explained by the possible misclassification of the indication of use (i.e. NHL versus CLL) or repeated infusions following an

interruption due to health problems of drug-induced toxicity, further investigations are warranted to exclude possible inappropriate utilization behaviors. The number of new users with only 1 or 2 administrations was negligible in both groups. Although these patients could be considered undertreated, the possible onset of serious adverse reactions (e.g. immune-related events) is likely to explain the suspension of the treatment. As for the average administered dose, a marked increase was observed after the first administration in the CLL group only, which was in line with the recommendations of use. [1]

Although other studies analyzed different inpatient populations and used different methodologies [20–22], we observed a comparable four-year survival of approximately 70% following the first rituximab administration. Using data from 12 German cancer registries, Pulte et al. [20] reported a 5-years survival of 80% since the first diagnosis of CLL, regardless of the use of rituximab or any other chemotherapy. A study from Jaime-Pérez et al. [21] reported a 5-year overall survival of patients diagnosed with NHL ranging from 63.8% (diffuse large B cell NHL) to 70.6% (follicular NHL).

Clinical trials on rituximab have shown controversial results due to the association of this drug with infections. [23] We observed that sepsis is the most frequently reported event following rituximab use. Table 3 showed 15/16 sepsis occurred in patients with NHL, which is not surprising, due to the more aggressive nature of this disease which often requires more intense chemotherapy, particularly in younger fit patients, compared to CLL.

However, the observed percentage of patients with at least one hospital diagnosis of infection during the first year of treatment, as well as the nature of the observed infectious events, appeared to be compatible with those reported in clinical trials. [4,24] The SmPC reports that 4% of patients treated with rituximab in monotherapy experienced severe infections. [1] Rituximab use is contraindicated in patients with preexisting active infections. In particular, cases of hepatitis B reactivation, including fulminant hepatitis with fatal outcome, have been reported with rituximab treatment [1,25], therefore screening for hepatitis B infection should be performed in all patients before initiating the treatment. Treatment with rituximab may also worsen the outcome of primary hepatitis B infection. With this respect, we observed two cases of hepatitis B infection recorded during the first year of treatment, though it is difficult to establish whether the infection was preexisting or not at the time of rituximab first administration. As for infectious events, given the exploratory nature of this study on the feasibility of linking RAD with HPS to perform pharmacoepidemiologic studies on infusive antineoplastics, the factors that might have predisposed patients to infectious complications were not collected so that we cannot establish the actual role of rituximab with respect to the occurrence of the observed events and whether they occurred in presence of neutropenic fever. Although assessing whether sepsis and the other infectious events observed could or could not be attributable to rituximab was not the aim of this descriptive study, it should be acknowledged that factors other than rituximab use, such as the advanced disease state, neutropenia, immunosuppression from other chemotherapy agents, might explain the occurrence of the infectious events observed. In particular, concomitant chemotherapy can have a significant quali-quantitative impact on the infectious events observed in rituximab treated patients. Moreover, patients who receive rituximab alone as induction therapy are likely to have significantly different survival and clinical characteristics compared to those who also receive concomitant chemotherapy. Although the information on concomitant chemotherapy was not available in the dataset authorized for the execution of this exploratory study, we can assume that approximatively all patients in the present study cohort received an induction combination therapy. Such an assumption is also supported by the high rate of serious adverse infectious events observed in this study. Conversely, a phase III randomized trial from Taverna and colleagues reported only 8-grade III-IV infectious events in 165 patients with follicular lymphoma exposed to maintenance therapy with rituximab monotherapy during 5 years follow-up. [26]

The main strength of this study was the linkage of two distinct sources of data routinely collected for administrative and hospital pharmacy management purposes. The two data sources provided complementary information on rituximab utilization in real-word setting and allowed to observe a large number of patients treated with rituximab for hematological indications during a long follow-up period. While HPS provided fundamental information on the actual indication of rituximab use and treatment line, which are not collected in administrative data, the integration with RAD containing health data (hospitalization, death, and health services' consumption), allowed to longitudinally follow patients treated with rituximab, even outside the University Hospital of Siena, providing information on patient's admissions to other Italian hospitals, or outpatient infusion center, vital status and emigration. The possibility of following patients even outside the hospital of Siena represents an important strength of this study. Although describing for the first time rituximab use, this study also has several limitations. First, HPS data was used for the first time for drug utilization research purposes and no validation studies are available. However, information retrieved from this source appeared to be consistent with clinical guidelines and treatment recommendations suggesting a high degree of reliability of the information recorded. [1,27,28] Another limitation of the study is represented by the lack of other important clinical information that is relevant to the description and evaluation of rituximab use, patient survival, occurrence and management of adverse infectious events in clinical practice. Indeed, information on tumor subtype (e.g. diffuse large B cell NHL versus follicular NHL), histology and stage, or patients' performance status, or antimicrobic drugs used during the hospital stay is not available in the data sources used for this study since this information is usually recorded in medical records. However, with our approach, we provided for the first time a general overview of the real-world utilization pattern of rituximab, adverse infectious events and the long-term survival in patients treated in a single Italian institution, which represents a novelty itself. Moreover, our experience set the stage for future studies in which the linkage of other existing intra-hospital sources of healthcare data (pathology registries, medical chart) will be able to be explored in order to retrieve clinically meaningful information which will allow to better evaluate the real-world utilization pattern of rituximab and also provide evidence on the occurrence and management of serious and potentially fatal adverse infectious events.

## Conclusions

Through the combined use of hospital pharmacy and administrative data, this study provided evidence on the real-world utilization of rituximab in a large adult patient population with NHL and CLL at UHS. The observed pattern of utilization was in line with treatment guidelines and current knowledge on rituximab safety profile The linkage of patient-level data from HPS and RAD obtained in this study, demonstrates the great potential for future pharmacoepidemiological studies on antineoplastic medications administered in the Italian hospital setting in Italy. Further studies are needed to explore the combined use of existing intra-hospital data sources other than hospital pharmacy (e.g. medical records, pathology registries). Considering the high costs of these drugs and the associated adverse reactions, monitoring the real-world utilization and appropriateness of use is of paramount importance both for patient's health and health care system sustainability.

## Supporting information

**S1 Table. Hospitalizations due to adverse infectious events recorded up to one year from the first rituximab administration.** The table shows the total number of hospitalizations due to adverse infectious events recorded up to one year from first rituximab administration

(events were also subdivided by Non-Hodgkin Lymphoma (NHL) and Chronic Lymphocytic Leukemia (CLL)). ICD-9CM codes about infectious events were reported.
(DOCX)

**S2 Table. Time-to-onset of adverse infectious events occurred up to one year from the first rituximab administration.** The table shows the time-to-onset of adverse infectious events that occurred up to one year from the first rituximab administration (both primary or secondary positions of hospital discharge records were considered). Meantime, median time, as the range days of the time-to-onset were reported.
(DOCX)

**S3 Table. Time-to-onset of adverse infectious events occurred up to one year from the first rituximab administration.** The table shows the time-to-onset of adverse infectious events occurred up to one year from the first rituximab administration (only primary positions of hospital discharge records). Meantime, median time, as the range days of the time-to-onset were reported.
(DOCX)

## Author Contributions

**Conceptualization:** Alessandro Barchielli, Sandra Donnini, Rosa Gini, Marina Ziche.

**Data curation:** Claudia Bartolini.

**Formal analysis:** Claudia Bartolini, Rosa Gini.

**Investigation:** Andrea Spini, Valentino Moscatelli.

**Methodology:** Giuseppe Roberto, Rosa Gini.

**Project administration:** Rosa Gini, Marina Ziche.

**Supervision:** Alessandro Barchielli, Davide Paoletti, Silvano Giorgi, Alberto Fabbri, Monica Bocchia, Sandra Donnini, Rosa Gini, Marina Ziche.

**Writing – original draft:** Giuseppe Roberto, Andrea Spini.

**Writing – review & editing:** Giuseppe Roberto, Andrea Spini, Sandra Donnini, Marina Ziche.

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
