## [Decision Letter · Decision Letter 0]

3 Dec 2019

PONE-D-19-30012

Patterns of drug use, survival and adverse infectious events in patients treated with rituximab at the University Hospital of Siena, Italy: exploring the feasibility of combining administrative and hospital-pharmacy data

PLOS ONE

Dear Dr Ziche,

Thank you for submitting your manuscript to PLOS ONE. After careful consideration, we feel that it has merit but does not fully meet PLOS ONE’s publication criteria as it currently stands. Therefore, we invite you to submit a revised version of the manuscript that addresses the points raised during the review process by both reviewers, experts in the field.

We would appreciate receiving your revised manuscript by Jan 17 2020 11:59PM. To enhance the reproducibility of your results, we recommend that if applicable you deposit your laboratory protocols in protocols.io, where a protocol can be assigned its own identifier (DOI) such that it can be cited independently in the future. For instructions see: http://journals.plos.org/plosone/s/submission-guidelines#loc-laboratory-protocols

We look forward to receiving your revised manuscript.

Kind regards,

Francesco Bertolini, MD, PhD

Academic Editor

PLOS ONE

Journal Requirements:

"Ethical Committees section “Area Vasta Sud-Est” in July 2016, code identifier:

ARSAOUS2016. The data were analyzed anonymously"

Please amend your current ethics statement to confirm that your named institutional review board or ethics committee specifically approved this study.

'The work was in part sponsored by AIRC 15443 (MZ).'

'The funders had no role in study design, data collection and analysis, decision to

publish, or preparation of the manuscript.'

Additional Editor Comments (if provided):

Reviewers' comments:

Reviewer's Responses to Questions

**Comments to the Author**

1. Is the manuscript technically sound, and do the data support the conclusions?

Reviewer #1: Yes

Reviewer #2: Yes

2. Has the statistical analysis been performed appropriately and rigorously? 

Reviewer #1: Yes

Reviewer #2: Yes

3. Have the authors made all data underlying the findings in their manuscript fully available?

Reviewer #1: Yes

Reviewer #2: Yes

4. Is the manuscript presented in an intelligible fashion and written in standard English?

Reviewer #1: Yes

Reviewer #2: Yes

5. Review Comments to the Author

Reviewer #1: The manuscript by the group of Marina Ziche gives a real picture of the appropriate administration of rituximab and its clinical effect in terms of overall survival in patients affected by non-Hodgkin's lymphoma and chronic lymphocytic leukemia with a focus on adverse infectious events related to the therapy.

The mansucript deals with an interesting, new retrospective approach: it looks at the Hospital Pharmacy of Siena and the Regional Administrative Database of Tuscany databases to extrapolate with a rigourous procedure the data regarding rituximab. The study is undoubtly interesting because it gives the possibilty to use also administrative databases to check the appropriatness of a drug administration and, above all, to have a real-world picture of the efficacy and toxicity of an important drug. This information is fundamental for many future studies, such as the pharmacoeconomic ones.

However, despite the efforts of the authors, there are some points that need to be clarified and better discussed:

1. as well stated in the introduction, rituximab is administered alone but also in combination with other chemotherapeutic drugs. Could the authors list these concomitant drugs and describe the survival effects in the subset of patients administered with these combinations? Furthermore, is there a difference in the averse infectious disease with those patients administered with rituximab alone? If these data could not be retrieved, please deeply discuss these issues into the discussion section of the text;

2. the authors listed the limitations of the study in the discusson section. The reviewer really appreciated this honest way to look at the shown data. However, could the authors include a list of antimicrobic drug that have been administered to the patients with infectious ADRs? these treatment were effective on rituximab-treated patients?

Finally, probably the short title is the ideal title for the manuscript. The reviewer suggests to adopt it.

Reviewer #2: in this interesting study the authors report a collimating cross between prescription, clinical data and information flows of the drug rituximab.

It would be intersteting to kow how may patients had a treatment with the use of Rituximab as a single drug and how many patients had a central venous catheter implanted. Moreover about the infectious event reported, should be useful to mantion the work by Taverna C et al (Rituximab Maintenance for a Maximum of 5 Years After Single-Agent Rituximab Induction in Follicular Lymphoma: Results of the Randomized Controlled Phase III Trial SAKK 35/03 J Clin Oncol. 2016 Feb 10;34(5):495-500) and comment the observed results in the paper submitted to PLOS

6. PLOS authors have the option to publish the peer review history of their article (what does this mean?). If published, this will include your full peer review and any attached files.

Reviewer #1: No

Reviewer #2: No

---

## [Author Response · Author response to Decision Letter 0]

11 Feb 2020

We appreciate the interest shown by the editor and the reviewers towards our manuscript. We are grateful for the comments raised by the reviewers which we believe helped us to better explain the limitations of the study and its actual scientific value. 

Please, find below our point by point responses to the reviewers’ comments.

Reviewer #1:

The manuscript by the group of Marina Ziche gives a real picture of the appropriate administration of rituximab and its clinical effect in terms of overall survival in patients affected by non-Hodgkin's lymphoma and chronic lymphocytic leukaemia with a focus on adverse infectious events related to the therapy.

The manuscript deals with an interesting, new retrospective approach: it looks at the Hospital Pharmacy of Siena and the Regional Administrative Database of Tuscany databases to extrapolate with a rigourous procedure the data regarding rituximab. The study is undoubtly interesting because it gives the possibility to use also administrative databases to check the appropriateness of a drug administration and, above all, to have a real-world picture of the efficacy and toxicity of an important drug. This information is fundamental for many future studies, such as the pharmacoeconomic ones.

However, despite the efforts of the authors, there are some points that need to be clarified and better discussed:

1. as well stated in the introduction, rituximab is administered alone but also in combination with other chemotherapeutic drugs. 

- Could the authors list these concomitant drugs and describe the survival effects in the subset of patients administered with these combinations? Furthermore, is there a difference in the averse infectious disease with those patients administered with rituximab alone? If these data could not be retrieved, please deeply discuss these issues into the discussion section of the text;

1) Unfortunately, we do not have the data needed to perform such additional analyses since the dataset extracted from Hospital Pharmacy of Siena for which we obtained permission to perform the present study does not contain information on concomitant chemotherapy. 

However, according to the authorized onco-hematologic indications and the experience at the University Hospital of Siena of our reference clinicians and co-authors, M. Bocchia and A. Fabbri, new users of rituximab who start on monotherapy are extremely rare (around 5% expected) and differs significantly in terms of clinical characteristics compared to those on combination therapy.

We thanks the referee for this comment since we acknowledge that the statement we reported in the introduction section about the possibility of using rituximab in combination with chemotherapy or in monotherapy was too general and possibly misleading. 

To better clarify this point, the introduction of the manuscript was modified as follow (pag. 3 line 57 of the revised manuscript):

“As for the haematological malignancies, with the exception of early stage follicular lymphoma, rituximab is considered the first-line treatment for patients with NHL and CLL. In newly treated patients rituximab is administered in combination with chemotherapy. Monotherapy with rituximab is indicated as maintenance therapy for patients who responded to induction therapy. Monotherapy is also indicated in patients with stage III-IV follicular lymphoma who are chemo-resistant or those in their second or subsequent relapse after chemotherapy.” 

We also added this sentence to the discussion section (line 253 – 261):

“In particular, concomitant chemotherapy can have a significant quali-quantitave impact on the infectious events observed in rituximab treated patients. Moreover, patients who receive rituximab alone as induction therapy are likely to have significantly different survival and clinical characteristics compared to those who also receive concomitant chemotherapy. Although information on concomitant chemotherapy was not available in the dataset authorized for the execution of this exploratory study, we can assume that approximatively all patients in the present study cohort received an induction combination therapy. Such assumption is also supported by the high rate of serious adverse infectious events observed in this study. Conversely, a phase III randomized trial from Taverna and colleagues reported only 8 grade III-IV infectious events in 165 patients with follicular lymphoma exposed to maintenance therapy with rituximab monotherapy during 5 years”

2. the authors listed the limitations of the study in the discussion section. The reviewer really appreciated this honest way to look at the shown data. However, could the authors include a list of antimicrobic drug that have been administered to the patients with infectious ADRs? these treatment were effective on rituximab-treated patients?

2) The point raised by the reviewer is definitely of high scientific interest. However, the data sources used for this study do not collect the information needed to answer this question. The database of the Hospital Pharmacy of Siena only provides information on infusive antineoplastics. Moreover, infectious events described in this study were retrieved from hospital discharge records collected in the regional administrative databases of Tuscany. The latter data source does not record neither drug utilization during inpatient stay nor clinical outcomes needed to assess antimicrobic therapy effectiveness. Access to medical records would be necessary to answer this question.

We added this point among the study limitations and future perspectives that are reported from line 286 to line 299in the discussion section.

3. Finally, probably the short title is the ideal title for the manuscript. The reviewer suggests to adopt it.

3) As suggested by the referee we adopted the following title:

“Real word evidence on rituximab utilization at the University Hospital of Siena, Italy: combining regional administrative and hospital-pharmacy data”

Reviewer #2: 

in this interesting study the authors report a collimating cross between prescription, clinical data and information flows of the drug rituximab. It would be intersteting to know how may patients had a treatment with the use of Rituximab as a single drug and how many patients had a central venous catheter implanted. 

4) As for the use of rituximab as a single drug, please see the answer 1 to the Referee #1.

As concern central venous catheter, it represents the standard of care at the University hospital of Siena, so that all patients in the study cohort are expected to have received central venous catheter implantation. 

According to referee suggestion, we searched administrative data where this event is expected to be recorded. However, no record of such procedure was found (ICDM9 code 38.97 - central venous catheter implanted). As confirmed by the local coding personnel, the cost of this procedure is negligible compared to the other cost items claimed by the hospital for the healthcare assistance of rituximab users so that coding priority of central venous catheter implantation is extremely low.

5. Moreover about the infectious event reported, should be useful to mantion the work by Taverna C et al (Rituximab Maintenance for a Maximum of 5 Years After Single-Agent Rituximab Induction in Follicular Lymphoma: Results of the Randomized Controlled Phase III Trial SAKK 35/03 J Clin Oncol. 2016 Feb 10;34(5):495-500) and comment the observed results in the paper submitted to PLOS

5) We thanks the referee for suggesting us this interesting reference which we cited in the text as follow (please see also answer 1 to the Referee #1):

“ ….we can assume that approximatively all patients in the present study cohort received an induction combination therapy. Such assumption is also supported by the high rate of serious adverse infectious events observed in this study. Conversely, a phase III randomized trial from Taverna and colleagues reported only 8 grade III-IV infectious events in 165 patients with follicular lymphoma exposed to maintenance therapy with rituximab monotherapy during 5 years”

---

## [Decision Letter · Decision Letter 1]

20 Feb 2020

Real word evidence on rituximab utilization: combining administrative and hospital-pharmacy data

PONE-D-19-30012R1

Dear Dr. Ziche,

We are pleased to inform you that your manuscript has been judged scientifically suitable for publication and will be formally accepted for publication once it complies with all outstanding technical requirements. As you can see, Reviewer #2 was unsatisfied with your asnwers, but I believe you did everything you could to answer her/his questions and that the manuscript deserves to be published.

With kind regards,

Francesco Bertolini, MD, PhD

Academic Editor

PLOS ONE

Additional Editor Comments (optional):

Reviewers' comments:

Reviewer's Responses to Questions

**Comments to the Author**

1. If the authors have adequately addressed your comments raised in a previous round of review and you feel that this manuscript is now acceptable for publication, you may indicate that here to bypass the “Comments to the Author” section, enter your conflict of interest statement in the “Confidential to Editor” section, and submit your "Accept" recommendation.

Reviewer #1: All comments have been addressed

Reviewer #2: All comments have been addressed

2. Is the manuscript technically sound, and do the data support the conclusions?

Reviewer #1: Yes

Reviewer #2: Yes

3. Has the statistical analysis been performed appropriately and rigorously? 

Reviewer #1: Yes

Reviewer #2: Yes

4. Have the authors made all data underlying the findings in their manuscript fully available?

Reviewer #1: Yes

Reviewer #2: Yes

5. Is the manuscript presented in an intelligible fashion and written in standard English?

Reviewer #1: Yes

Reviewer #2: Yes

6. Review Comments to the Author

Reviewer #1: The authors provided sensible answer to the raised comments and consequently amended the manuscript.

Reviewer #2: the authors responded adequately to the reviewers' comments but the lack of the requested data makes the paper unacceptable for publication

7. PLOS authors have the option to publish the peer review history of their article (what does this mean?). If published, this will include your full peer review and any attached files.

Reviewer #1: No

Reviewer #2: No

---

## [Editor Report · Acceptance letter]

25 Feb 2020

PONE-D-19-30012R1 

Real word evidence on rituximab utilization: combining administrative and hospital-pharmacy data 

Dear Dr. Ziche:

I am pleased to inform you that your manuscript has been deemed suitable for publication in PLOS ONE. Congratulations! Your manuscript is now with our production department. 

With kind regards,

on behalf of

Dr. Francesco Bertolini 

Academic Editor

PLOS ONE